# Revisiting the Domestication Process of African *Vigna* Species (Fabaceae): Background, Perspectives and Challenges

**DOI:** 10.3390/plants11040532

**Published:** 2022-02-16

**Authors:** Davide Panzeri, Werther Guidi Nissim, Massimo Labra, Fabrizio Grassi

**Affiliations:** 1Department of Biotechnology and Bioscience, University of Milan-Bicocca, Piazza della Scienza 2, 20126 Milano, Italy; werther.guidi.nissim@gmail.com (W.G.N.); massimo.labra@unimib.it (M.L.); 2Department of Agriculture, Food, Environment and Forestry (DAGRI), University of Florence, Viale delle Idee 30, 50019 Sesto Fiorentino, Italy

**Keywords:** *Vigna* genus, introgression, hybridisation, phylogeny, *de novo* domestication, feralisation, bioactive compounds

## Abstract

Legumes are one of the most economically important and biodiverse families in plants recognised as the basis to develop functional foods. Among these, the *Vigna* genus stands out as a good representative because of its relatively recent African origin as well as its outstanding potential. Africa is a great biodiversity centre in which a great number of species are spread, but only three of them, *Vigna unguiculata*, *Vigna subterranea* and *Vigna vexillata*, were successfully domesticated. This review aims at analysing and valorising these species by considering the perspective of human activity and what effects it exerts. For each species, we revised the origin history and gave a focus on where, when and how many times domestication occurred. We provided a brief summary of bioactive compounds naturally occurring in these species that are fundamental for human wellbeing. The great number of wild lineages is a key point to improve landraces since the domestication process caused a loss of gene diversity. Their genomes hide a precious gene pool yet mostly unexplored, and genes lost during human activity can be recovered from the wild lineages and reintroduced in cultivated forms through modern technologies. Finally, we describe how all this information is game-changing to the design of future crops by domesticating *de novo*.

## 1. Introduction

Legumes (Fabaceae) are considered one of the most important families of plants for human nutrition, especially considering the rapid growth rate of the world population [1]. However, almost all the efforts and resources invested in agriculture during the last century were focused on improving the yield, resistance and quality of a few specific staple crops. Neglected landraces are regarded as having interesting potential, and recent studies have demonstrated that some wild legumes can be an important target to develop modern functional foods because they possess various bioactive molecules that interact positively with human health [2,3,4,5]. Among these, members of the *Vigna* genus show a growing social and economic importance in several African regions, especially where the local population is not able to afford animal proteins [6,7,8]. Their seeds are rich in essential amino acids and contain a high concentration of minerals, lipids and vitamins [9,10].

The genus *Vigna* (Savi, 1824), which belongs to the tribe Phaseoleae of the family Fabaceae, includes over 100 species [11] distributed in the tropical and subtropical areas of the world [12] grouped in five subgenera: *Vigna*, *Ceratotropis*, *Plectotropis*, *Lasiosporon* and *Haydonia* [13,14,15]. Phylogenetic findings propose the age of split between *Phaseolus* and *Vigna* genera at about 8–10 million years (Mya) and the age of split between *Ceratotropis* and *Vigna* subgenera at about 3–4 Mya [13,14,15,16,17], but the genetic relationships between subgenera are particularly complex and far from being completely solved. Although most domesticated or semi-domesticated species are distributed in Asia, the greatest diversity of the *Vigna* genus is located in Sub-Saharan Africa [14,18]. *Vigna* subgenus, distributed in Africa, includes about 40 wild and 2 domesticated species, namely cowpea (also called black-eyed peas, chawli and kunde) (*Vigna unguiculata* L.) and Bambara groundnut (*V. subterranea* L.) [19] while *Ceratotropis* (Piper) Verdc., distributed in Asia, contains 21 wild and 7 domesticated species used widely for food and forage, namely mungbean or green gram (*V. radiata* L. Wilczek), black gram (*V. mungo* L. Hepper), moth bean (*V. aconitifolia* Jacq. Maréchal), rice bean (*V. umbellata* Thunb. Ohwi and Ohashi), adzuki bean (*V. angularis* L. Ohwi and Ohashi), creole bean (*V. reflexo-pilosa* Hayata), jungli bean (*V. trilobata* L. Verdc.). [15,20,21,22]. Moreover, three species belonging to *Plectrotropis* (Schumach.) are distributed in Africa, including tuber cowpea (*V. vexillata* L.) [23]. Most of the African *Vigna* germplasm is based on wild plants and neglected or underutilized landraces, and many of these lineages are declining with a high risk of extinction. The recovery of wild accessions and research devoted to the phylogeny of the genus is therefore essential to prevent genetic erosion and the loss of *Vigna* diversity.

Plant domestication is widely recognised as an accelerated evolutionary process driven by a synergistic impact of human and natural selection, occurring in geographically restricted areas from wild progenitors. In legumes, the main modification is the loss of seed pod dehiscence or shattering [24,25]. The split at the dorsal and ventral sutures of the dry pod and successive release of the seeds occurs due to the desiccation of lignified cells in the pods [26]. The shattering habit is related to environmental aridity and persists in many varieties of domesticated *Vigna* species, thereby determining severe yield losses [27,28]. Additional implementations in *Vigna* domesticated species include an increase in seed or fruit size, change in seed colour, loss of seed dormancy, apical dominance and change in flowering timing [29,30,31,32,33]. These modifications were inherited more or less effectively in the various vine species currently cultivated, and this is the basis of the agrobiodiversity of this genus.

Generally, the current existing crops show lower resistance to biotic and abiotic stress compared to wild relatives, and often they have reached their full yield. The selection of desirable traits and breeding processes to improve crop productivity have caused the depletion of diversity and the increase in the frequency of deleterious genetic variants that are fixed in the genomes of crops [34,35,36]. These constraints have a serious impact on agriculture, limiting the possibility to grow such crops under more extreme environmental conditions. Thanks to this residual genetic diversity and also to studies performed on *Vigna* species, most of the accessions are well adapted to a wide range of extreme environmental conditions, such as sandy beaches, arid lands and wetlands, harbouring tolerance and resistance genes towards biotic and abiotic stresses. These genetic traits are used for developing new stress-tolerant crops [37,38,39,40,41,42,43]. By contrast, less is known about the effects of domestication on the nutritional value of seeds [7] even if recent studies have reported that cultivated legumes show a lower carotenoid and protein content in seeds compared with the wild relatives [44,45]. Where, when and how many times the domestication process of African *Vigna* crops occurred continues to be debated among researchers. Although archaeological remains of *Vigna* indicate that the domestication process in Africa was started recently compared to other field crops [46,47]. Modern evolutionary models proposed for other crops suggest that the predomestication phase may have lasted several thousands of years [48,49]. Generally, the centres of origin are also recognized as centres of diversity, and thus these areas require special precautionary measures of conservation [50]. Although for many crops the single-origin model is usually the most parsimonious, the hypothesis that provides multiple origins starting from independent founder lineages seems well suited for the crops of *Vigna* originated in Africa [51,52]. Moreover, despite whether and to what extent introgression influences the domestication process is still underexplored, some studies already show that gene flow between cowpea and its wild relatives may occur. Pervasive introgression can also intensify the feralisation process, promoting the crops to return to a wild environment and causing serious problems for the conservation of biodiversity [53].

In this review, we re-examine the available scientific information on the domestication process of three African *Vigna* crops and discuss the future perspectives and challenges in the light of modern technologies in the time of climate change and new parading of conservative agriculture strategies. Another crucial point in exploring natural biodiversity is not only a matter of sustainability but also a matter of human health. A balanced diet gives extreme benefits to people’s wellbeing by properly assuming the correct amount of micro and macro nutrients as well as useful, healthy bioactive compounds. Finding and characterising these compounds is an ambitious challenge for researchers thus we briefly summarise the bioactivity of some compounds, and we discuss how human activity and breeding has impacted the variability of molecules.

Although recent genetic studies have led to a deeper understanding of these crops, the continuation of investigating the domestication process through a multidisciplinary approach which includes genomic, transcriptomic, metabolomic and epigenomic analyses is needed to highlight the wide agronomic opportunities related to these species. Moreover, recent techniques of gene editing have opened new and crucial ways to redesign modern crops because traditional genetic improvement is generally limited by the cross-compatibility between species. Thus, because the *de novo* domestication process may represent a turn toward more modern and sustainable agriculture, further efforts are needed to explore the genome diversity of wild germplasm.

## 2. *Vigna unguiculata* (L.) Walp.

*V. unguiculata*, which was considered an orphan crop for several decades, has recently become one of the most important legumes in the world. Its name derives from Latin and means “with a small claw”, referring to the size of the claw of the petals [54] or commonly named as “cowpea” because of its use as fodder for cows [55] and black-eyed pea/bean for the black hilum. This crop is largely cultivated, especially in semiarid regions of Africa and Asia where other crops fail to grow [10]. Currently, on a global scale, about 15 million hectares are dedicated to *V. unguiculata,* with an annual production of 7 million Mg and an average yield of 0.6 Mg ha^−1^ [56]. The most interesting environmental traits of this species are represented by the generalized low agrochemical input requirements. In fact, this crop shows relatively high adaptation to drought, especially in comparison to other legumes [57] and can fix up to 200 kg N ha^−1^ [58] with a positive soil N balance of up to 92 kg ha^−1^ [59]. Nevertheless, several abiotic and biotic constraints (i.e., low soil fertility, pests, diseases, parasitic weeds, and nematodes) limit the yield [43,60,61]. Moreover, low productivity is often associated with the use of traditional and unimproved varieties, still widely cultivated in Africa [62]. This crop has a fundamental role in human nutrition, showing seeds rich in proteins and essential amino acids (i.e., tryptophan and lysine), carbohydrates, folic acid and minerals. Recent studies carried on a large sampling have shown high variability in protein and mineral concentrations, suggesting that some lineages could be potential sources of genes useful to produce new varieties [63,64,65,66].

The high number of wild subspecies found exclusively in Africa strongly supports the idea of an African origin. However, intraspecies phylogeny remains far from being completely elucidated [67]. The centre of origin of the species is probably located in the southernmost regions of Africa, where most subspecies are found and where most genetic diversity could be still hidden [68]. Several taxonomic revisions based on morphological and molecular traits permitted to identify 10 perennial and 1 annual subspecies, the latter split into two varieties: ssp. *unguiculata* var. *unguiculata* (domesticated cowpea) and ssp. *unguiculata* var. *spontanea* (Schweinf.) Pasquet., also known as subsp. *dekindtiana* sensu Verdcourt non Harms [69,70,71,72,73,74,75,76]. Besides the domesticated cowpea, the *dekindtiana* group includes some obligate short-day wild forms, well adapted to arid environments. While the var. *spontanea* grows especially around cultivated fields and roadsides, and it is recognized as the progenitor of domesticated cowpea [75,77,78,79], the subspecies *alba*, *pubescens*, *tenuis*, *stenophylla* and *dekindtiana* are perennial plants [75,76]. The development of new molecular tools to discriminate among wild, weedy, and cultivated accessions is considered a modern and fundamental target, particularly needed for disentangling the complex taxonomic relationships among subspecies and to discriminate between true wild plants and ferals.

Although little is known about the domestication process, some scientists have hypothesized that ancient cowpea progenitors, such as the modern wild forms, were adapted to dry habitats and grew spontaneously south of the Sahara Desert [80]. These plants were gathered, cultivated and dispersed by men near the villages, but they were unsuited for cultivation. Although they did not show high yield, the wild lineages were spread in the humid zones thanks to their pods that remained closed for the humid atmosphere. Through several generations of cultivation, new mutants have arisen, showing interesting domestic traits, including resistance to shattering. Subsequently, humans have selected and helped spread these landraces by exchange and trade activities. Since the oldest archaeological records of domesticated forms found in central Ghana are dated around 1500 BC, the domestication process likely started before that period (Figure 1) [47,81]. However, the precise origin is widely debated, and two independent domestication centres in West and East Africa are proposed by different authors [68,74,79,82,83,84,85,86].

Morphological and DNA markers support the idea that domestication occurred only once, but analyses on whole genomes provide evidence for more independent domestication events in Africa and diversification events out of Africa [51,87]. Analyses of genetic variability are generally applied to identify the origin of species and the groups of accessions that show high variability in certain geographic areas and are interpreted as the most ancient populations. Although cowpea from West Africa showed a high genetic variability [88], cultivated accessions grown in East and West Africa were shown to be most closely related to the respective local wild lineages [52,89], thereby indicating that domestication could have occurred in both regions. Outside Africa, cultivated cowpea was exposed to different ecological conditions, including new biotic and abiotic stresses that probably have contributed to shaping the genetic structure of landraces. When cowpea moved through Asian regions (especially in Thailand, China, the Philippines and India), it encountered environments with more humidity and less brightness where the drying of pots and grains was hindered. Some accessions were selected for the use of the immature pods to produce a peculiar form of vegetable called yardlong bean (*V. unguiculata* ssp. *unguiculata* cv. *sesquipedalis*) [51,90,91]. Although Chinese accessions show lower genetic diversity compared to African cowpea, signals of genetic bottlenecks lead to the conclusion that a limited number of relatively recent selection events occurred;however, where the selection process arose is still unknown [92]. Moreover, other cultivar groups (e.g., ‘Textilis’, ‘Biflora’ or ‘Cylindrica’, ‘Melanophthalmus’) are classified by morphological traits [75,93]. Still, additional genomic analyses should be performed to confirm the genetic relationships and understand how and where these accessions originated [67,85,88,94,95].

## 3. *Vigna subterranea* (L.) Verdc.

*Vigna subterranea*, also named Bambara groundnut, is an indigenous African grain legume. Its common name derives from the groundnut (*Arachis hypogaea* L.) due to the hypogean pods’ growth, whereas the “Bambara” name is derived from a Malian tribe [96]. Despite its potential in terms of nutritional value and resistance to biotic and abiotic stresses [97,98], Bambara is cultivated mainly in small farms or in families as a subsistence crop [99], and naturally grows in semi-arid regions in Africa. Regarding the origin of the species itself, the domesticated or semi-domesticated *Vigna subterranea* var. *subterranea* was most likely generated from its wild counterpart *Vigna subterranea* var. *spontanea* using both morphological and isozyme data [100,101].

The origin of this species is hypothesised to be in Mali, in the Timbuktu region [102], but the precise centre of origin is still unknown. In fact, there is no evidence of wild lineages in Mali [103]. Dalziel, Begemann and Goli [104,105,106] analysed a lot of morphologic traits such as seed morphology, seed pattern diversity and other diversity indices (number of days to maturity, pod length, number of stems per plant and internode length). They found that the most diversity is located in an area that spans from Jos Plateau and Yola Adamawa (Nigeria) to Garoua (Cameroon). Somta and Olukolu [107,108] evaluated the phylogeography of several accessions spread in Africa. The markers used (i.e., SSR and DaRT) showed a cluster with higher diversity in the area between Nigeria and Cameroon. The authors confirm the area of origin while suggesting a possible subsequent introduction of Western domesticated accessions in East Africa (Figure 1). In contrast, Aliyu et al. [97], in an overview of the past two decades of genetic diversity analysis, also proposed that the Southern African region could constitute a divergent time-spaced domestication event. However, the authors suggest that these hypotheses need further examination.

In terms of genetic diversity, Bambara has a peculiar behaviour. In fact, many authors studied Bambara’s genetics with different techniques to clarify how wide the genetic pool is and how homogeneous the single landraces are. Molosiwa et al. [109] evaluated genetic distances between 24 landraces with phenotypic and genetic markers (i.e., SSR and DaRT). The main results report that landraces are different to each other, suggesting the existence of great allelic diversity among the various populations. At the same time, though, single landraces tend to be very homogeneous, and in three generations of inbreeding became pure lines. This is due mainly to its self-pollinating nature [110] but also small farmers, who, by breeding the same landraces, also acted as selection drivers [111,112]. Molosiwa [113] selected 12 SSR markers and 5 Bambara accessions to evaluate the potential for creating pure lines, finding that these accessions at the second cycle of selection completely have lost the heterozygosity.

All these findings suggest that Bambara has incredible genetic potential. The genetic screening through the different lineages and the consequent discovery of peculiar sites of interest could be the basis for an improvement of crop programs. Moreover, the use of pure lines in agriculture is fundamental not only for the optimisation and standardisation of agricultural practices but also for the development of breeding programs. Currently, to the best of our knowledge, there are no reports of ongoing improvement or breeding programs for this species. The extremely wide pool of wild and domesticated accessions can be used to create ideal crops that can better withstand climate change as well as being able to grow with low agronomic inputs.

## 4. *Vigna vexillata* (L.) A. Rich.

Widely distributed in Africa, Asia, America and Australia, *V. vexillata* (Zombi pea) is one the least known and underutilized *Vigna* crops. Likewise, *V. unguiculata*, Zombi pea shows a high morphological diversity probably determined by geological, ecological, climatic and anthropomorphic constraints that also determined exceptional patterns of genetic variability [19,71]. Eight varieties including *vexillata*, *angustifolia*, *ovata*, *dolichomena*, *yunnanensis*, *plurifora*, *lobatifloria* and *macrosperma* are recognized [12,19,23,114,115]. Var. *macrosperma* shows typical traits associated with domestication syndrome such as bush-like habit, early flowering and higher seed yield [116,117]. Moreover, loss of seed dormancy and various degrees of pod shattering were detected in different crop accessions while the wild seeds remained intrinsically dormant [118,119]. Several authors reported that two forms were domesticated independently (i.e., seed type and tuber type), and some evidence lines suggest that the seed type was domesticated in Sudan, whereas the tuber type was domesticated in India (Figure 1) [120,121,122,123,124]. However, molecular analyses were performed on a limited number of accessions and loci [124], and the phylogenetic intra-specific delimitation has resulted in much more complexity than that of other *Vigna* crops [125]. Thus, modern genomic analyses are needed to resolve the genetic relationships and confirm the origin of the two forms.

Several studies have also shown that the Zombi pea is the result of a long adaptation process to different environmental stress, including acid, alkaline, saline, drought and wet soils [115,117,126,127,128]. Moreover, since some accessions were found to be resistant to different viral diseases and parasite insects, widely recognized as major pests of cowpea, this species is an important harbour of resistances to various biotic stresses, particularly useful to improve modern *Vigna* crops [129,130,131,132,133,134].

## 5. Healthy Natural Compounds for Designing Sustainable Crops

The process of domestication was selected during the early millennia due to all the characteristics that made a species very productive or easier to harvest. Nowadays, a lot of crops varieties that have a great yield and high contents of macronutrients exist, such as carbohydrates or proteins. However, bioactive compounds that are naturally present in the *Vigna* genus were never taken into account. In a world where the main concern is no more denutrition but instead malnutrition, the adoption of crops with high-value nutraceutical compounds becomes a challenge for the next generation. The *Vigna* genus is a great source of small proteins and secondary metabolites with nutraceutical roles in everyday diet.

Often agricultural practices themselves could stimulate the production of these compounds, such as hydric stress or no tillage with cover crops fields. However, they could not be sufficient to enhance the output of bioactive molecules. In this perspective, *de novo* domestication programs should consider these compounds to develop future healthy crops. In the next paragraph, we listed and discussed some of these molecules based on the nutraceutical activity they exert against three great world concerns.

### 5.1. Antioxidant and Anti-Inflammatory Activity

Nowadays, inflammation and oxidative stress are becoming great concerns due to detrimental effects on human health, and diet is a powerful way to protect cells from the rise of reactive oxygen species (ROS) as well as inflammatory processes. In this view, seeds of cowpea contain different phenols, and other pigments present in the seed coat [135] are able to promote antioxidant action; among these, quercetin and flavonols are very well represented [136]. Different works [137,138] demonstrated a clear correlation between antioxidant properties and the colour of the seed coat in different accessions of *Vigna vexillata* and *Vigna subterranea*. Sowndhararajan and Leu [139,140] identified that *Vigna vexillata* extracts three molecules with strong antioxidant properties. Daidzein, abscisic acid and quercetin were highly active and displayed a pivotal capacity to deny the inflammation pathway.

Studies performed on protein extracts of *Vigna subterranea* suggest that different protein fractions exert crucial properties against ROS relevant in cellular metabolism [141]. Furthermore, a review by Quan et al. [142] summarised how polyphenols and proteins naturally interact, providing a higher antioxidant and anti-inflammatory capacity as a result.

The presence of antioxidant compounds is clearly a good starting point for the bioprospecting approach. The research could start from accessions already studied and kept in germplasm banks, with the aim of breeding the most promising ones (e.g., more colourful, thicker coat or better nutrient profile) with domesticated landraces to create variants that are, at the same time, easy to cultivate but with the most interesting characteristics found from the available natural pool. In addition, this could lead to new experiments to understand better the synergic role of the phenolic fraction with bioactive proteins.

### 5.2. Anti-Tumor Compounds

Concerning the anti-cancer activity, Bowman–Birk inhibitors (BBI), present exclusively in the Fabaceae family and some cereals [143], have proven anticancer effects [144,145]. Panzeri et al. [146] demonstrated that aqueous extracts from boiled seeds containing BBI are, as expected, effective against some colorectal cancer cell lines, but the healthy line was not hit by the treatment. Mehdad et al. [147] proved its activity on breast cancer lines, and they were the first to discover a potential intracellular target, the proteasome 20S. Furthermore, they demonstrated cytostatic activity and increased apoptosis in cancer lines, but BBI was ineffective on the healthy mammary epithelial line. It is important to underline that this protein is kept by evolution due to its defensive role; in fact, it inhibits herbivores’ digestion by interacting negatively with trypsin and chymotrypsin. Preliminary results obtained via the alignment of sequences downloaded from genebanks (NCBI) showed a high variability of BBI gene in some cowpea accessions, confirming the greatness of natural biodiversity. Unfortunately, little is known about the impact of domestication on the variability of the BBI gene. The domestication process can have acted as a strong constraint causing a bottleneck in the gene pool and reducing the variability of genes and exchange of alleles between cultivated and wild accessions. However, the exploration of haplotypes by sequencing several accessions is needed to verify the effective impact of human activity on gene diversity. Moreover, methods of ancestral sequence reconstruction (ASR) based on phylogenetic inference can predict the existence of stable, soluble, and active variants of proteins. The comparison of the structure of modern proteins with the corresponding ancestral intermediates can highlight functionally important substitutions within proteins and consequently drive the protein engineers to design variants that confer novel or more efficient activities (Figure 2). While different case studies are discussed in the literature where ancestral reconstructions were applied in eukaryotes, few instances are available in plants. Since ASR can be used to explore the remote evolutionary past as well as to investigate molecular evolution on shorter timescales, we argue that the proteins expressed in different genera of legumes are particularly well suited for ancestral reconstruction studies.

Phenolic acids (e.g., gallic acid, ferulic acid, caffeic acid and chlorogenic acid) and flavonols (catechins, kaempferols and quercetins) are groups of molecules that are very active against cancer. Teixera-Guedes et al. [148] found some of these molecules in the phenolic fraction of cowpea sprouts. Sprouting is an alternative method to consume food, especially seeds, grains and pulses. As a matter of fact, sprouting refers activating the metabolism of the dormant seeds and this way, complex reserve molecules are degraded into simpler ones, releasing other molecules and secondary metabolites [149]. The authors demonstrated at first the efficacy of the extracts against CRC cell lines; then combined it with 5-Fluorouracil (5-FU). This drug is potent but is susceptible to the occurrence of resistance by the tumour mass [150]. Among all these compounds, quercetin is one of the most common, was found to be the main representative of extracts and is well known to be active against different cancer lines [151,152,153].

The capacity to exert different kinds of bioactivities appoints phenolics and small proteins as very potential phyto complexes with an extreme wideness of possible applications. In this paragraph, the fact that extracts can be much more effective than single drugs is underlined. The use of a mixture of bioactive compounds in addition to the chosen drug could help in the treatment of many diseases.

### 5.3. Anti Hypercholesterolemic

One of the major world concerns is the role of the diet for healthy living. In particular, the main problem is malnutrition, 1.9 billion adults are overweight, and 452 million are underweight [154]. These numbers are going to increase during the next few years, so a healthy diet must become a worldwide topic. One way to prevent obesity is to find food or molecules that can lower LDL cholesterol concentration or production. Legumes are known to have a good nutritional profile and possess some interesting anti hypercholesterolemic capacities. For example, in the work of Tan et al. [155], *Vigna subterranea* was the object of study to create a powdered drink mix. The authors managed to characterise the extracts and proved the ability to lower the total cholesterol content in a population of rats. The observed effects were comparable to those given by the commercial drug simvastatin, demonstrating a potential commercial formulation usable in everyday life. In addition, Bambara powder fat content was lower than the soybean, while it had more proteins. Regarding *Vigna unguiculata*, Kanetro [156] studied the hypocholesterolemic feature of protein extracts from the sprouts. The tests were performed on rats that mimicked a diabetic condition. This kind of extract established the potential of *Vigna unguiculata* in fighting high cholesterol concentrations. *Vigna unguiculata* was also studied in rabbit models by Janeesh and Abraham [157]. Rabbits received a rich fraction polyphenols and flavonoids extracted from the leaves that showed antioxidant capacity, hypolipidemic and anti-atherogenic properties in ill animals. The road opened by the studies reported here is encouraging and already tending to practical applications usable worldwide by combining the natural nutritive features to bioactive compounds present in the seeds.

Although many important bioactivities are reported in this paragraph, the actual knowledge is still incomplete. Small proteins and polyphenols were objects of these studies, and their versatility in terms of the panel of bioactivities exerted was highlighted and valorised over and over. A topic that we would like to stress more and encourage research on is the variability of seed coat colours. In fact, human activity has selected a wide range of shapes, textures and pigments in coats (including eye shapes and sizes), allowing us to clearly distinguish seeds of domestication accessions from unattractive seeds of wild lineages. The seed colours are correlated to the presence of tannins and flavonoids [158,159], and phenolic profiles showed that seed coats contain up to 10 times more flavonoids if compared to whole seeds [160]. The seed coat pattern is a fundamental aspect of consumer preference, but in different regions, only some patterns are preferred. On the other hand, local landraces contain a great variability of colours, selected through centuries by human activity, but often this richness remains undervalued [135,137,161]. Our suggestion is to use this kind of information to correlate the colours of seed coats with the proper chemical characterisation regarding previously mentioned bioactivities. Moreover, modern experimental planes should include wild accessions/species and underused landraces because these mostly unexplored taxa could hide important micronutrients. Finally, we underline that the introduction of new dishes based on a mix of seeds that show different colours could be a new way to assimilate a great variety of nutrients into the diet.

## 6. Introgression and Feralisation Processes

Through the domestication process, one or more populations that showed desirable traits are selected by humans producing new independent lineages. Farmers have strongly influenced the survival of these cultivated lineages that continued to diverge from wild ancestors because they were affected by different selective pressures. However, crops and their wild relatives can exchange genetic information spontaneously or through human activity. Generally, wild relatives of legumes show undesirable traits, but their genomes can hide a precious gene pool that is mostly untapped that can be recovered and reintroduced in cultivated forms.

Introgression of useful genes remains a fundamental way to improve the cultivar [162], and successful crosses mediated by humans were acquired, especially in cowpea [163]. Although domesticated cowpea is known as an inbred crop, outcrossing is reported suggesting that frequency and distance can vary depending on the environment, climate, subspecies, genotype and insect involved [73,164,165]. In *Vigna unguiculata,* spontaneous introgressive events between wild perennial subspecies of the *dekindtiana* group, including accessions of var. *spontanea*, are widely described observing different morphological traits [71,84,166,167,168]. Molecular analysis using AFLP [74] and internal transcribed spacers [53,169,170] have confirmed the natural propensity to hybridisation between subspecies and have revealed intricate intraspecies phylogenetic relationships. Intragenomic 5S rRNA repeat unit heterogeneity was interpreted as the consequence of extensive hybridisation events [170], and recently, plastid DNA sequences have confirmed chloroplast capture events [76].

In recent decades, several researchers have tried to produce introgressive lineages obtaining interesting results and showing that the most important gene pool for breeding programs could be harboured in wild subspecies. Intraspecific hybrids obtained crossing ssp. *unguiculata* with ssp. *pubescens* and cv. *sesquipedalis* with ssp. *tenuis* have shown vigorous growth and partial fertility [171,172,173]. Some authors attributed the incomplete success to chromosomal disturbances that ensue in endosperms and embryos during early seed development when crosses between wild perennial accessions and domesticated cowpeas are performed [174]. However, different accessions showed diverse propensity to hybridize, and a recent study suggests that temperature and humidity also have a prevalent role in increasing the success of hybridisation [175]. A wild lineage of cowpea (TVNu-1158) collected in the Republic of Congo showed resistance to *Aphis craccivora,* surviving long after infestation [176], and was successively crossed with cowpea to produce new lineages [177]. Moreover, resistance to *Maruca vitrata* was observed in the wild lineage of ssp. *dekindtiana* (Tvnu 863) from Zimbabwe andresistance to *Clavigralla tomentosicollis* was observed in ssp. *dekindtiana* (TVnu 151) from Ghana; however, literature about their use to produce new cultivars is missing [178,179].

Limited information is available about the intraspecific introgression of *V. subterranea* and *V. vexillata*. The success of the artificial cross of Bambara is constrained by scarce pollen viability, the small size of the flower and the reduced stigma–anther separation, which improves the transfer of pollen to the stigma but at the same time complicate the emasculation process [180,181,182,183]. However, F1 and F2 lines were obtained by crossing *Vigna subterranea* var. *spontanea* (Harms) Pasquet and *Vigna subterranean* var. *subterranea* (L.) Verdc. varieties, allowing us to identify that the main morphological traits to distinguish the two forms (internode length and stems per plant) are regulated by relatively limited numbers of genes [184]. Intraspecific introgression success was also obtained by James and Lawn [185] who crossed African and Australian accessions of *V. vexillata* with the aim to explain the resistance to mottle carmovirus (CPMoV). Recently, modern hybridisation techniques were applied to cross var. *macrosperma* cultivated and wild accessions obtaining encouraging results [186,187]. Unfortunately, scarce findings are achieved by interspecific hybridisation. Differently from Asian taxa, where the compatibility was confirmed in different studies, the African taxa show a cross incompatibility barrier that has so far prevented the introgression of useful genes (e.g., *V. vexillata* × *V. unguiculata*) [168,188,189,190].

In recent years, advances in sequencing technologies have allowed the generation of a large number of genomic resources that, if combined with approaches that estimate the rate of gene flow, enable us to detect which lineages are prone to hybridisation. Screening the level of introgression already existing in nature is an important opportunity that can help us to obtain advanced information useful in breeding activity. For example, natural hybrid zones harbour genetic variance and, pervasive and occasional introgressive events are identified in several crops such as kiwi, common bean, soybean, sunflower and grape [191,192,193,194,195,196]. Differently from neutral introgression, which could be lost during successions of generations, adaptive introgression events are maintained by selection, and foreign gene variants introduced by gene flow can increase the fitness of receiver populations as observed in potato, rice and millet [197,198,199].

African *Vigna* species have a potential for introgression that today remains mostly unexplored. *V. unguiculata* and *V. vexillata* show an elevated number of wild lineages that probably have diverged well before the Pleistocene due to climate changes [71,125]. Several subspecies are adapted to different environments, and Padulosi and Ng [68] proposed that the southernmost region of Africa is presumably the origin center for *V. unguiculata* where most subspecies grow, while Pasquet [71] indicated that some lineages from Namibia to Zimbabwe are the result of spontaneous introgression events. However, genomic studies are needed to confirm these hypotheses, including in the analyses of populations spread at the margins of species distribution that could hide local adaptation to extreme conditions. Principal component analysis, Bayesian clustering methods (e.g., NEWHYBRIDS, STRUCTURE and ADMIXTURE) and divergence statistics such as FST are used to explore patterns of divergence in *Vigna* species, but they manifest shortness to provide the effective migration rate. To overcome this limitation, different probabilistic approaches recently developed are able to identify recent and ancient signals of introgression such as tree-based methods (e.g., Treemix), coalescent-with-introgression simulations (e.g., MSci model implemented in BPP), composite-likelihood test (e.g., VolcanoFinder), site frequency spectrum to explicitly model migrations (e.g., ∂a∂i), gene genealogies (e.g., Twisst) and ABBA–BABA statistics [200,201,202,203,204]. Moreover, only some genomic regions could be involved in gene flow, and thus introgression might be localised in specific chromosomes [205]. Since alleles shared through incomplete lineage sorting remain complex to distinguish from alleles shared through introgression and none of the measures described above is without simplifying assumptions, we suggest that different methods should be applied to ascertain the origin of introgression.

Although introgression from wild to crops has important economic consequences and many attempts are made to understand the evolutionary dynamics, in recent years, attention to the gene flow from crop to wild is rapidly increasing. Introgression of domesticated alleles can stimulate the evolution of weeds or increase the risk of extinction of wild populations with dramatic evolutionary consequences, as demonstrated in several annual and perennial plants [206,207,208,209,210,211]. Moreover, under specific circumstances, the spread of ferals escaped from cultivation and adapted to wild environments can hardly be contained. Although several authors consider feralisation the opposite process of domestication, few population genomics studies show how these genetic changes occurred in plants [212]. Some authors show that multiple de-domestication events have occurred in rice, highlighting that some crops are exceptionally prone to feralisation [213,214]. The introgression process is probably improved when the wild forms grow along the road margins, villages and fields where domesticated forms are cultivated. To date, few studies have investigated the introgression effects on the wild populations of African *Vigna* species. Some researchers have proposed that alleles from cowpea may be incorporated into wild forms especially improved by their cohabitation, replacing the original alleles and making new lineages well adapted to wild environments [215]. A molecular study based on analysis of isozyme loci showed that outcrossing rates in West Africa range from 1% to 9.5%, confirming possibilities of gene flow from domesticated cowpea to var. *spontanea* [216]. The distinction between feral and truly wild lineages is ever more complicated because introgression produces fertile offspring and the small seed-size typical of wild forms is dominant to large seed size [76,217]. Moreover, var. *spontanea* is represented by both annual and perennial plants, and it is acknowledged that while the annual and inbred habit is an adaptive strategy in dry and warm environments (e.g., in warm tropical savannas), perennials tend to grow in mountainous regions where the environment is often cooler and wetter [76]. Annual inbred plants produce more seeds and show a competitive advantage on perennials when they are sympatric in environments. Although few data about introgression are available, we cannot exclude that perennial outcrossed subspecies can be fertilized by cowpea pollen, and consequently, domesticated alleles can be introgressed. Moreover, feralisation can involve adaptive changes in genes related to flowering timing, dormancy and metabolic pathways, which are also unknown. Therefore, several aspects of the feralisation process, including the ability to spread domesticated alleles accross long distances by seeds and pollen through mammals or birds and the predisposition to invade territories where perennial subspecies grow, should be further explored.

## 7. Domestication-Related Traits and *De Novo* Domestication

As described by Darwin [218], most plants subjected to intensive domestication have lost the ability to survive in the wild environment for more than a few generations. Traits selected by humans allow us to clearly distinguish a domesticated plant from its wild progenitor, and several studies were recently proposed to highlight the genes at the base of these changes. In recent years, modern genomic techniques were applied to *Vigna* germplasm, accelerating research activity and opening new avenues to identify domestication-related traits [33,161,219,220,221].

Among the main domesticated traits in legumes, the loss of pod shattering and increase of organ size are most relevant for breeding. In cowpea, two main quantitative trait loci (QTLs) were identified for pod shattering, whereas QTLs identified for seed weight, leaf length, leaf width and pod length were located in the same region, suggesting a potential pleiotropy that controls the organ size [177]. Lonardi et al., 2019 [222] managed to obtain the entire genome sequence in order to analyse and identify the eventual putative syntelog for organ gigantism. They found a region containing a cluster of QTLs controlling pod length, seed size, leaf length and leaf width (CPodl8, CSw8, CLl8, CLw8). Similar results were also observed in *V. vexillata* where the main domestication traits, including seed size, pod size and leaf size, were controlled only by one or a major QTL and some minor QTLs [33,90,124]. More complex is the control of seed dormancy, which is generally managed by water-impermeable layers of cells of the seed coat. In yardlong bean, a vegetable crop that has experienced divergent domestication from cowpea, six QTLs were detected for seed dormancy-related traits [90]. The seed coat pattern is an essential trait in cowpea, intensely selected by human preferences that change in the different areas of Africa. For example, pigmentation displays high variability of colours, including varied eye shapes and sizes. Recent studies show that the colour and position of the pigmentation can be defined by expression patterns, and some genes that encode for proteins involved in the flavonoid biosynthesis pathway were identified [161]. Moreover, phenotypic observations show that a lack of pigment in flowers is often correlated with a lack of pigment in the seed coat, and a gene was recently proposed to have a dual function in cowpea controlling the colour in both organs [177]. As observed for several species, the flower was involved intensely in the domestication process, and it has a fundamental trait that allows us to distinguish domesticated cowpea from their wild relatives. Recently, innovative studies focused on exploring the genetic basis of floral scent. A group of five O-methyltransferase genes involved in the biosynthesis of melatonin and located within the floral scent QTL region was identified [221]. Melatonin is recognized as an essential molecule in several plants used to interact with pollinators. Flowering timing undoubtedly plays a key role in plant adaptation and diffusion of crops because several agronomic traits such as grain quality, plant growth and plant height are directly influenced by this characteristic [223]. However, how the domestication process has affected the timing of flowering in legumes is unclear [224]. Flowering timing is a complex trait generally regulated by genetic networks. While in *Arabidopsis thaliana* L., the existence of up to 80 loci [225] was shown, in a cowpea genome-wide association study (GWAS) seven reliable SNPs were revealed that explained phenotypic variance [220]. Important agronomic implications are expected because the candidate genes could be transferred by hybridisation in crops. Early flowering accessions can mature earlier, avoiding periods of drought stress, whereas late-flowering accessions can mature later and extend the vegetative period, thereby increasing biomass production.

It is widely recognized that the study of domestication-related traits is a fundamental step that enables us to understand how to design ideal crops for the future. Throughout the process of domestication and successive breeding phases, the genetic diversity of crops was significantly reduced, and this homogeneity is becoming a serious threat. The increase of disease and inability of adaptation to environmental changes that consequently cause an increased use of pesticides and water with a severe impact on the environment are the main issues that affect the sustainability of modern agriculture. Fernie and Yan [226] emphasized that wild species contain less deleterious allelic variants than their crops, and Smykal et al. [227], in a recent review, reported that modern cultivars have lower levels of key vitamins and micronutrients, suggesting that several wild and semi-wild African species should be *de novo* domesticated.

Unfortunately, few studies of re-domestication are available, but recent advances in gene editing combined with the decryption of pan-genomes are opening new perspectives of manipulation of genes for the creation of modern crops [228,229]. Gene editing is used to modify the function of genes already existing, incorporate new genes and delete short or large DNA fragments [230,231]. For instance, undesirable traits can be reduced or removed by intervening in genes that regulate the content of secondary metabolites, accelerating the process of domestication. Otherwise, the life cycle of cowpea could be shifted coming back from annual to perennial, as occurred for *Triticum aestivum* L. [232]. Perennial cowpea crop would show deep roots, higher water and nutrients efficiency and would not need to be sown every year.

Modern techniques such as CRISPR/Cas9 are applied successfully in several staple food crops. In *Oryza sativa* L., mutations on three yield-related genes have produced more and larger grains and erect panicles [233], whereas, in *Solanum pimpinellifolium* L., eight genes were targeted improving architectural traits, day-length insensitivity, the size and shape of fruits and content of vitamins [234,235]. Moreover, the CRISPR-Cas9 system was also used in cowpea to disrupt the symbiotic nitrogen fixation by the modification of a symbiosis receptor-like kinase (SYMRK) gene, thereby demonstrating that gene editing can be applied to the *Vigna* genus [236]. However, this technique requires that the genome is sequenced to identify the ortholog gene that controls the domestication trait [237].

African *Vigna* species are an ideal group of plants on which to apply gene-editing techniques and to produce modern crops. A great number of wild species, besides showing resistance to pests and diseases and having high nutritional values, are well adapted to diverse environmental conditions [9]. Only Angola, with 28 native *Vigna* species documented, is recognized as one of the most important sources of germplasm in the world [238]. *V. monantha* occurs in permanently dry conditions [42], whereas *V. marina* and *V. luteola* grow well in saline lands [9]. In particular, seedlings of *V. marina* can survive for several weeks in flooded conditions and high NaCl concentration [39], accumulating high levels of salt in leaves, roots and stem [41].

However, few farmers currently use these plants because of low yield and strong pod-shattering behaviour, which requires high labour during the harvest. Adaptation to extreme environments often involves multiple genes, whereas domestication-related traits are due to mutations of a single locus that affects the loss of a function. Previously reported domestication-related traits in *Vigna* seem to be controlled by a restricted number of QTLs. Thus, introducing domestication-related mutations into wild species might be preferred rather than modifying multiple genes related to complex adaptation traits. For example, the first steps of re-domestication were achieved by Takahashi [4], starting from the accessions of *Vigna stipulacea* (Lam.) Kuntze originated in Asia. The authors obtained one mutant with reduced pod shattering and three mutants with reduced seed dormancy, characterizing the respective SNPs in the candidate genes. *V. stipulacea* was selected for their fast growth, edible seeds andbroad resistance to pests and diseases. Thus, *de novo* crops can be designed to preserve several traits that nature has selected in millions of years. Moreover, in the next few years, the pan-genomes of several economically important crops will be available. The investigation by sequencing multiple individuals, including wild and domesticated accessions, will allow us to acquire full knowledge of variations at the genome level. Since it is widely accepted that the use of few reference genomes is limiting, the pan-genome of the *Vigna* species should be achieved in a short time [239,240]. Consequently, given the large diversity of wild *Vigna* germplasm spread in Africa and the modern techniques of gene editing, great margins of genetic improvement are expected in the near future.

## Figures and Tables

**Figure 1 plants-11-00532-f001:**
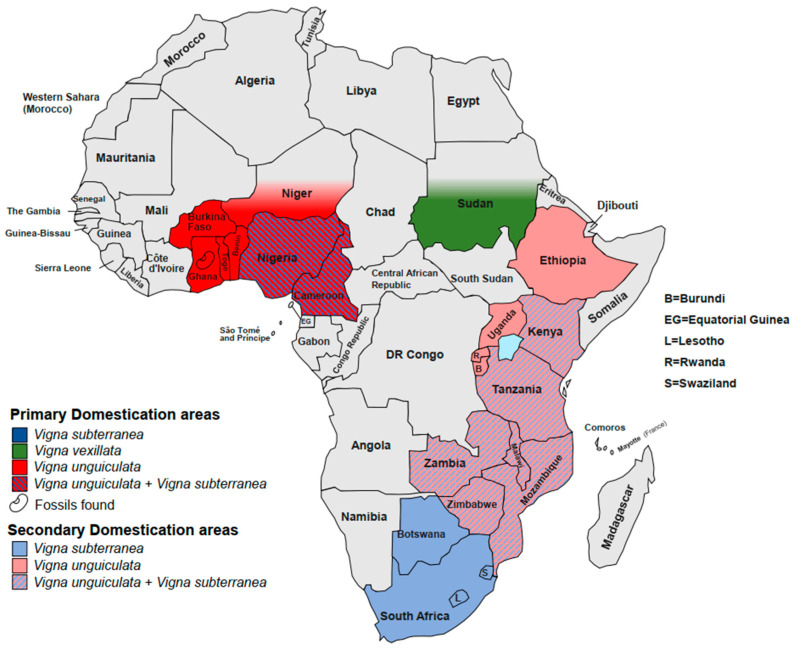
Primary and secondary domestication sites in Africa.

**Figure 2 plants-11-00532-f002:**
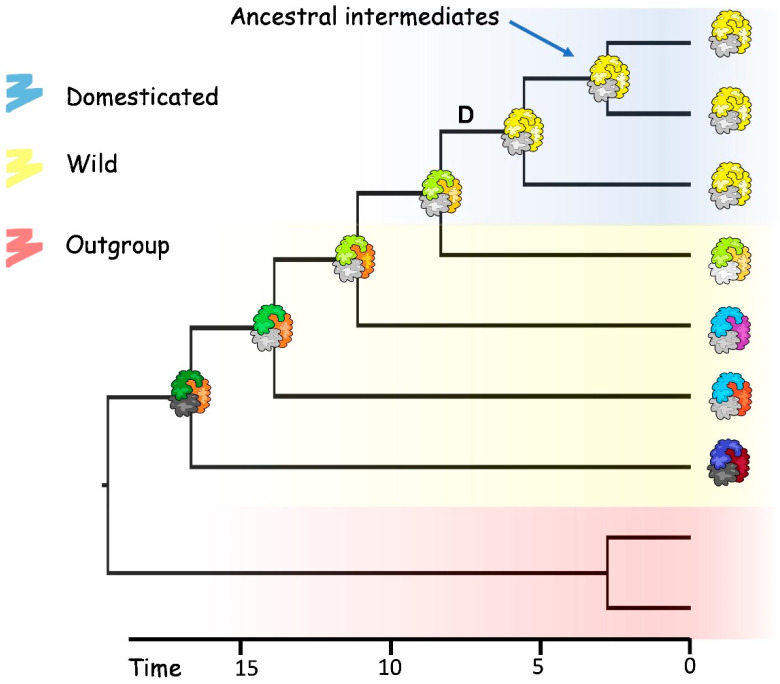
An example of reconstruction of putative ancestral intermediates by inferring the phylogenetic relationship between modern homologs. ASR studies can explore biodiversity to infer the historical evolution of natural proteins. Statistical models of amino acid substitution can be applied to calculate the sequences at internal nodes. Although domestication (D) has produced bottlenecks and reduced the genetic variability, ASR analyses can be applied to wild crop relatives.

## Data Availability

Not applicable.

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
