# Peer review of "Revisiting the Domestication Process of African Vigna Species (Fabaceae): Background, Perspectives and Challenges"

_plants, 2022, doi:10.3390/plants11040532_

Round 1

Reviewer 1 Report

The work presented by Panzeri et al. is interesting. It appears to be well documented. In this report, the authors explored the center of origin, domestication, introgression of domestication-related traits, and feralization of three African Vigna crops, namely, Vigna unguiculata, Vigna subterranea and Vigna vexillate. The authors also provided an insight into the use of state-of-the-art genome editing techniques with a view to further improving the Vigna crops. This manuscript is worth publishing and can be accepted for publication in ‘Plants’ after addressing the following points.

  1. There is a conundrum about the origin of the Chinese yardlong bean Vigna unguiculata (L.) Walp. ssp. unguiculata-gr. Sesquipedalis. Some studies reported that it is originated in China. However, the authors mentioned on page 5, line # 184-186 that “…Cowpea spread in Asia was selected for use of the immature pods to produce a peculiar form of vegetable called yard-long bean…”. Therefore, more discussion is required to resolve this conflict. At least, it should be mentioned in which part of Asia the African Cowpea was domesticated.
  2. Cross-incompatibility is a major problem for Vigna According to the authors, as the unavailability of the genome is restricting researchers from improving Vigna crops by using the CRISPR-cas9 technique, what should be done to improve these crops.
  3. The genome of cowpea is available! Lorandi et al. (2019) already assembled and generated 519 Mb genome data of a single haplotype inbred line of cowpea called IT97K-499-35. They discussed the gene families available in the cowpea genome. The authors also identified the candidate gene for multiple organ gigantism from previously published QTL data. That information must be included in this manuscript.
  4. There is a type on page 10 line #431 “…obtained by 210 James and Lawn”.
  5. It would be nice if the “African Vigna species: a source of healthy molecules” section could be uncoupled from this manuscript and published as a separate manuscript. It does not seem fit in this manuscript where domestication and further improvement of African Vigna crops are the main purposes. Besides, no domestication events ever targeted better chemical compounds and associated health benefits.

Author Response

We thank the reviewer for the precious comments and suggestions provided. Here is the list of our improvements and answers:

There is a conundrum about the origin of the Chinese yardlong bean Vigna unguiculata (L.) Walp. ssp. unguiculata-gr. Sesquipedalis. Some studies reported that it is originated in China. However, the authors mentioned on page 5, line # 184-186 that “…Cowpea spread in Asia was selected for use of the immature pods to produce a peculiar form of vegetable called yard-long bean…”. Therefore, more discussion is required to resolve this conflict. At least, it should be mentioned in which part of Asia the African Cowpea was domesticated.

We thank the reviewer for the comment. We improved the discussion regarding the origin of yardlong bean and added the putative geographical areas of Asian domestication. (Lines 188-195, Page 7)

Cross-incompatibility is a major problem for Vigna According to the authors, as the unavailability of the genome is restricting researchers from improving Vigna crops by using the CRISPR-cas9 technique, what should be done to improve these crops.

As described in the paragraph "Introgression and feralisation process", the wide exploration of the levels of introgression in nature, including wild populations in the analyses, could be a way to overcome the problem of cross-incompatibility in relatively short times. On the other hand, new technologies of genome editing are now available and some findings have been shown. CRISPR-Cas9 or similar techniques will allow us to produce the food of the future and now we can imagine only some putative applications. The number of genomes completely sequenced is growing in all crops and in the next few years new accessions will be available. However, to acquire full knowledge of genetic diversity and to gain a full understanding of genomic variations, the pan-genome of Vigna should be planned. Although these arguments are treated along the manuscript we have added some sentences at the end of the text to explain these concepts. (Lines 661-665, Page 17)

The genome of cowpea is available! Lorandi  et al. (2019) already assembled and generated 519 Mb genome data of a single haplotype inbred line of cowpea called IT97K-499-35. They discussed the gene families available in the cowpea genome. The authors also identified the candidate gene for multiple organ gigantism from previously published QTL data. That information must be included in this manuscript.

We thank the reviewer for this suggestion. We added all these important information about the availability of the whole genome and organ gigantism in the text with the reference you provided us. (Lines 571-575, Page 15)

There is a type on page 10 line #431 “…obtained by 210 James and Lawn”.

We removed the typo. (Line 481, Page 13)

It would be nice if the “African Vigna species: a source of healthy molecules” section could be uncoupled from this manuscript and published as a separate manuscript. It does not seem fit in this manuscript where domestication and further improvement of African Vigna crops are the main purposes. Besides, no domestication events ever targeted better chemical compounds and associated health benefits.

We thank the reviewer for the suggestion. This paragraph was discussed a lot during the writing of this manuscript but we think it should be kept. The main reason is that we wanted to highlight a series of properties that are very important for human health but have been overlooked during domestication. Furthermore we believe that could be a converging point between these disciplines that rarely communicate with each other and could give new points of view and research impulses.

Following your suggestion then, we tried to reconfigure the introduction of the paragraph in order to make the read smoother and more focused on the domestication importance. We changed the title of the paragraph and we also slightly modified the bioactivity sections and deleted unnecessary or off-topic sentences. (Pages 8-12)

Reviewer 2 Report

‘Revisiting the domestication process of African Vigna species (Fabaceae): background, perspectives and challenges’ provides a very interesting synthesis about three crop species of the genus Vigna in Africa and their respective wild relatives.

This is a well written manuscript, very well organized and easy to read, even considering the very diverse information included therein.

Besides some minor suggestions (see below), I consider that the work can be publish and I think that it will be a good contribution to the state-of-the-art about these crops and to identifying future areas of research relevant to crop improvement in Vigna genus.

I have only a few suggestions:

Figure 1 - No reference is made, nor in the text neither in the figure, to the mixed patterns corresponding to combined situations (primary and secondary domestication areas). Also, they are not easily recognised (perhaps widest stripes would be more evident).

Ln. 126 – ‘the shape of stalks on the petals’ – please check this. I think that it refers to the size (small) of the claw of the petals (not to the shape of stalks).

Ln. 133 – the work Hall 2012 is not in References 

Ln. 174 - ‘Morphological and molecular data support the idea that domestication occurred only once, but genomic analyses provided evidence for more independent domestication….’ - In this context what is meant with molecular data,on the one hand, and genomic analyses, on the other hand.

Ln. 185 - ‘…to produce a peculiar form of vegetable called yard long bean or sesquipedalis’ – sesquipedalis is a common name? Perhaps use the scientific name (Vigna unguiculata subsp. sesquipedalis).

yard long or yardlong (see line 524)?

Ln. 188 - (e.g.,Textilis, Biflora or Cylindrica, Melanophthalmus) - Single inverted commas should be used around cultivar names (e.g.,’Textilis’, ‘Biflora’ or ‘Cylindrica’, ‘Melanophthalmus’)

Ln. 269 - 'we list 3 characteristics' – writte '3' in full (three)

Ln. 273 – ‘and the Diet is a powerful way’ – Diet with lower case

Ln. 398 – ‘Spontaneous introgressive events between wild perennial subspecies of dekindtiana group, including accessions of var. spontanea, are widely described observing different morphological traits’ – At the beginning of the text is not mentioned that it refers to Vigna unguiculata.

Ln. 427 – ‘crossing spontanea and subterranea subspecies’ – these taxa (subspecies) exist? As far as I know only varieties were described (var. subterranea and var. spontanea (Harms) Pasquet). Please consider including the authorities in the species scientific names.

Ln. 431 - by 210 James and Lawn [185] – 210?

Ln. 451 – ‘V. unguiculata and V. vexillata show an elevated number of wild subspecies’ - in the case of V. vexillate I suppose that only varieties were described (not subspecies)

Ln. 517 – ‘In cowpea two main QTLs were’ – QTL written in full the first time is referred.

Ln. 610 - 'For research articles with several authors, a short paragraph specifying their individual contributions must be provided.' – delete this sentence

Check References – scientific names are not properly written (e.g. Vigna Marina Subsp. Oblonga should be Vigna marina subsp. oblonga).

Author Response

We thank the reviewer for the comments and suggestion given. Here is a list of comments ando our answers or modifications.

Figure 1 - No reference is made, nor in the text neither in the figure, to the mixed patterns corresponding to combined situations (primary and secondary domestication areas). Also, they are not easily recognised (perhaps widest stripes would be more evident).

We thank the reviewer for this comment. We provided a better clarification of the image according to your suggestion by changing the colour pattern plus a description in the legend. In addition, we uploaded and updated the PDF version of the map which is vectorial and with higher definition while in the text there is only a screenshot. (Graphical Abstract and Figure 1)

Ln. 126 – ‘the shape of stalks on the petals’ – please check this. I think that it refers to the size (small) of the claw of the petals (not to the shape of stalks). 

We updated the text with your correction. (Line 129, Page 6)

Ln. 133 – the work Hall 2012 is not in References 

We deleted the Hall 2012 reference because it was a citation used for an older version of the manuscript. (Line 136, Page 6)

Ln. 174 - ‘Morphological and molecular data support the idea that domestication occurred only once, but genomic analyses provided evidence for more independent domestication….’ - In this context what is meant with molecular data,on the one hand, and genomic analyses, on the other hand.

We modified the sentence in order to make it clearer for the reader. (Line 177-178, Page 7)

Ln. 185 - ‘…to produce a peculiar form of vegetable called yard long bean or sesquipedalis’ – sesquipedalis is a common name? Perhaps use the scientific name (Vigna unguiculata subsp. sesquipedalis).

yard long or yardlong (see line 524)?

We updated and formatted all the names. (Line 191-192, Page 7 and Line 579, Page 14)

Ln. 188 - (e.g.,Textilis, Biflora or Cylindrica, Melanophthalmus) - Single inverted commas should be used around cultivar names (e.g.,’Textilis’, ‘Biflora’ or ‘Cylindrica’, ‘Melanophthalmus’)

We added the inverted commas as you suggested. (Lines 199-200, Page 7)

Ln. 269 - 'we list 3 characteristics' – writte '3' in full (three)

Fixed (Line 287, Page 9)

Ln. 273 – ‘and the Diet is a powerful way’ – Diet with lower case

Done (Line 199, Page 9)

Ln. 398 – ‘Spontaneous introgressive events between wild perennial subspecies of dekindtiana group, including accessions of var. spontanea, are widely described observing different morphological traits’ – At the beginning of the text is not mentioned that it refers to Vigna unguiculata.

We provided the specification of the species we talked about. (Line 431, Page 12)

Ln. 427 – ‘crossing spontanea and subterranea subspecies’ – these taxa (subspecies) exist? As far as I know only varieties were described (var. subterranea and var. spontanea (Harms) Pasquet). Please consider including the authorities in the species scientific names.

Thanks for the comment. We checked thoroughly and we provided a correct identification of the varieties with the proper authorities. (Lines 476-478, Page 12)

Ln. 431 - by 210 James and Lawn [185] – 210?

It was a typo we lost during the final revision. It referred to an old citation numeration. (Line 481, Page 12)

Ln. 451 – ‘V. unguiculata and V. vexillata show an elevated number of wild subspecies’ - in the case of V. vexillate I suppose that only varieties were described (not subspecies)

Thanks for the comment, we followed your suggestion by changing “subspecies” to “lineages”. (Line 501, Page 14)

Ln. 517 – ‘In cowpea two main QTLs were’ – QTL written in full the first time is referred.

We provided the full name for QTL. (Lines 568-569, Page 15)

Ln. 610 - 'For research articles with several authors, a short paragraph specifying their individual contributions must be provided.' – delete this sentence

Done. (Lines 669-670, Page 17)

Check References – scientific names are not properly written (e.g. Vigna Marina Subsp. Oblonga should be Vigna marina subsp. oblonga).

Every scientific name should be correct now. (Pages 17-26)